# Dependence of Induced Biological Damage on the Energy Distribution and Intensity of Clinical Intra-Operative Radiotherapy Electron Beams

**DOI:** 10.3390/ijms241310816

**Published:** 2023-06-28

**Authors:** Rafael Colmenares, Rebeca Carrión-Marchante, M. Elena Martín, Laura Salinas Muñoz, María Laura García-Bermejo, Juan C. Oller, Antonio Muñoz, Francisco Blanco, Jaime Rosado, Ana I. Lozano, Sofía Álvarez, Feliciano García-Vicente, Gustavo García

**Affiliations:** 1Servicio de Radiofísica, IRYCIS-Hospital Universitario Ramón y Cajal, Carretera de Colmenar Viejo km 9100, 28034 Madrid, Spain; feliciano.garcia@salud.madrid.org; 2Grupo de Aptámeros, Departamento de Bioquímica-Investigación, IRYCIS-Hospital Universitario Ramón y Carretera de Colmenar Viejo km 9100, 28034 Madrid, Spain; rebecacm93@gmail.com (R.C.-M.); m.elena.martin@hrc.es (M.E.M.); 3Biomarkers and Therapeutic Targets Group, IRYCIS, RedinREN, Hospital Universitario Ramón y Cajal, Carretera de Colmenar km 9100, 28034 Madrid, Spain; laurasalinas04@gmail.com (L.S.M.); marialaura.garcia@salud.madrid.org (M.L.G.-B.); 4Centro de Investigaciones Energéticas, Medioambientales y Tecnológicas—CIEMAT, 28040 Madrid, Spain; jc.oller@ciemat.es (J.C.O.); antonio.roldan@ciemat.es (A.M.); 5Departamento de Estructura de la Materia, Física Térmica y Electrónica e IPARCOS, Universidad Complutense de Madrid, 28040 Madrid, Spain; pacobr@fis.ucm.es (F.B.); jaime_ros@fis.ucm.es (J.R.); 6Instituto de Física Fundamental, Consejo Superior de Investigaciones Científicas, 28006 Madrid, Spain; anita_ilm@iff.csic.es (A.I.L.); sofalvar@ucm.es (S.Á.); g.garcia@csic.es (G.G.); 7Centre for Medical Radiation Physics, University of Wollongong, Wollongong, NSW 2522, Australia

**Keywords:** high-energy radiotherapy, fast electrons, electron transport, Monte Carlo method, cell survival, radiation effects, HaCaT cells

## Abstract

The survival fraction of epithelial HaCaT cells was analysed to assess the biological damage caused by intraoperative radiotherapy electron beams with varying energy spectra and intensities. These conditions were achieved by irradiating the cells at different depths in water using nominal 6 MeV electron beams while consistently delivering a dose of 5 Gy to the cell layer. Furthermore, a Monte Carlo simulation of the entire irradiation procedure was performed to evaluate the molecular damage in terms of molecular dissociations induced by the radiation. A significant agreement was found between the molecular damage predicted by the simulation and the damage derived from the analysis of the survival fraction. In both cases, a linear relationship was evident, indicating a clear tendency for increased damage as the averaged incident electron energy and intensity decreased for a constant absorbed dose, lowering the dose rate. This trend suggests that the radiation may have a more pronounced impact on surrounding healthy tissues than initially anticipated. However, it is crucial to conduct additional experiments with different target geometries to confirm this tendency and quantify the extent of this effect.

## 1. Introduction

More than 50% of all patients diagnosed with cancer will require radiotherapy as a component of their oncologic care at some point during their disease course [1]. Although historically, most treatments have been delivered using photons, in some cases, electrons are preferred due to their physical characteristics, mainly their lower penetration in tissue. In particular, intraoperative electron radiation therapy (IOeRT) [2] was developed more than half a century ago. This approach combines surgical resection and high-energy electron irradiation to the tumour bed while protecting healthy tissues by removing them from the radiation field. IOeRT has demonstrated its efficacy in a wide variety of anatomical sites [3]. These localized treatment approaches aim not only to achieve local tumour control but also to preserve the patient’s quality of life to the greatest extent possible. Side effects, especially in locations, such as colorectal [4] or head and neck [5,6,7], are the counterpoint of radiotherapy treatments.

Absorbed dose [8], defined as the energy deposited in the matter by ionizing radiation per mass unit, is the reference magnitude for radiotherapy applications, including treatment prescription and evaluation. This magnitude, macroscopic and deterministic, correlates well with results through extensive clinical experience [9]. However, by definition, it cannot consider differences in the pattern of interactions at the molecular level and, thus, possible differences in the damage induced by different particles or energies, assuming the same absorbed dose [10,11,12]. There is no general method to account for radiation quality in clinical applications. In radiation therapy and protection practice, it is customary to assume that all electron and photon radiations have the same radiobiological efficacy (RBE) [13,14], regardless of their energy. However, Linear Energy Transfer (LET) [8] varies significantly with the kinetic energy of electrons [15], as it does for all charged particles. It is important to note that LET exhibits a sharp increase near the track-ends, where the energy has substantially decreased. These electrons can be considered the most critical in terms of its biological effects [16,17]. Consequently, the relative fraction of the absorbed dose due to low energy electrons can be related to the local biological effectiveness of the radiation [18]. Both high and low initial energy electrons have a relatively high LET at the end of their trajectories. However, a larger fraction of the absorbed dose deposited by low initial energy electrons occurs in this high-LET region [19,20]. Accordingly, some authors have suggested using RBE calculation models [20,21] based on the fraction of the absorbed dose due to electrons of energies lower than a few keV. This approach can be seen as a macroscopic consequence of the behaviour of DNA molecules with radiation.

These low-energy electrons, produced as secondary particles by any radiation beam, especially those with energy below 1 keV, are mainly responsible for inducing biological damage [16,19,22,23,24]. The distribution of stochastic interactions, known as track structure, depends on the type of particle and energy. Measurements at the molecular scale are difficult, if not impossible, so Monte Carlo Track Structure (MCTS) codes are the most powerful tools for calculating radiation–matter interactions and, in some cases, estimating their biological effects [16,19,25]. To this end, these codes track each particle, both primary and secondary, down to energies close to a few eV or lower. It should be noted that the uncertainty increases along with the process stages: physical, chemical and biological. In addition, the physics associated with electron interactions below 100 eV, especially in condensed media, such as water, remains an area of research [26]. 

DNA damage has historically been associated with double-strand breaks (DSBs) [27,28], as the production of these lesions has been observed to be approximately linear with the absorbed dose [29,30]. However, the correlation between the number of these lesions at the molecular level and a higher-level biological effect remains to be determined. Today, complex DSB and non-DSB clusters receive much more attention, and Monte Carlo track structure simulations have consistently shown that the proportion of these complex and clustered lesions increases as the electron energy decreases [31,32,33]. There is ample theoretical and experimental evidence accumulated over the years showing that this type of damage is more difficult to repair than isolated DSBs, due, at least in part, to the reduced ability of cells to repair DSB clusters through the NHEJ pathway. Therefore, the efficient production of clusters rather than individual DNA lesions by ionizing radiation is widely considered a critical initiating event for mutagenesis, genomic instability, and cell death [34,35,36,37,38,39,40]. However, some discrepancies in the available data indicate that different factors may be involved in determining the repair pathway of choice [41,42]. 

Consequently, there are still unresolved fundamental questions in radiation biology, such as the physical interactions leading to DNA damage and the relationship between damage occurring at the molecular level and its consequences at the cellular level. Semi-empirical models, such as the linear-quadratic model [43], have been employed to estimate the cellular response to irradiation. Other models based on DNA repair processes have also been developed in recent years [44].

The radiobiological properties of clinical electron beams have been studied from different scientific perspectives and objectives. The published experimental RBE values for Linac electron beams (energy > 4 MeV) are in the range of 0.9–1 [45,46,47] compared to a ^60^Co photon beam. These experimental values have been obtained at different energies, target cells, biological assays and procedures, and even at different dose levels. For these energies, the value usually assigned to electrons is the same as that of MV-photons. On the other hand, for Auger emitters with electron emission mostly <1 keV, RBE values of 5 or higher are recommended [48]. In all these experiments, the electron beam was evaluated as a static energetic entity with no variation in the spectrum since they were performed at a single depth. 

The changes in the energy spectrum of the incident electrons also lead to changes in the interaction pattern, which could cause different biological damage depending on the depth. In the case of protons, this effect is more prominent due to their pronounced Bragg peak, and there is extensive scientific documentation of this effect. In the case of electrons, although there is no macroscopic Bragg peak but rather a more diffuse behaviour due to their lower mass, the ultimate reason that could lead to the same effect is the continuous loss of energy as they advance in the biological medium.

The effect of depth in the RBE for MeV electron beams was debated and controversial in the late 1960s and 1970s. Through different studies, some authors found a more or less pronounced increase in biological damage with depth [49,50]. However, other authors did not measure this effect [51,52,53,54]. These experiments commonly utilized electron beams with energies of 23–35 MeV, but this energy range for electrons is currently in clinical disuse. In addition, there was significant diversity in the methodology, irradiating animals in some cases and cells in others.

Despite the lack of scientific consensus, the interest in these studies was declining, probably due to the decreasing use of electrons as therapeutic beams. However, interest in this field has been renewed over time as MCTS codes have been developed and the theoretical key role of low-energy electrons has been emphasized.

A connection between the cellular scale and radiation quality was established through microdosimetric models [55], such as the Theory of Dual Radiation Action (TDRA) [56] and the Kinetic Microdosimetric Model (MKM) [57]. These models calculate RBE values from microdosimetric variables obtained with MCTS codes considering the radiation quality. MKM is based on the linear-quadratic model, is tissue-specific, and works by defining the survival level, while TDRA is more generic and is defined for a given dose level. TDRA has been widely used to estimate the RBE of different types of particles [58,59,60], including electrons [61,62]. Chattaraj et al. [63] calculated the RBE values based on TDRA and MKM with a FLUKA-MCTS code [64] for different therapeutic electron beams (6, 12 and 18 MeV). For the 6 MeV beam, the change in the RBE_TDRA value between the depths of 0.2 × R50 (R50 is the depth at which 50% of the maximum dose is reached) and 1.2 × R50 was around 18%. However, for the same beam when using the MKM model, RBE hardly varies in that depth range. 

In a recent paper [65] using GATE (a Geant4-based tool) and the MKM model, no significant differences were observed in the microdosimetric estimator of biological damage (the dose-weighted linear energy) with varying depths in water for the 20, 100, and 300 MeV electron beams. However, for the 5 MeV beam, the same methodology and calculation showed an increase in biological damage with depth. That is, the ratio of *more-damaging* electrons (≤50 keV) [66] passing through a depth to those with higher energy increases continuously in a specific way depending on the initial spectrum. Experimental results are crucial to establish whether the biological damage of polyenergetic electron beams is depth-dependent or whether this effect occurs below a specific nominal energy. Experimental studies have been scarce, with methodological differences and involving different energies. There is no consensus on these issues, especially for electron beams with energies below 10 MeV. This energy range, 4–9 MeV, is commonly used in radiotherapy treatments. In this context, this work aims to study the variations in the biological damage of HaCat cells due to the spectrum change with depth in water for a 6MeV electron beam. In addition, the physical phase of the beam has been simulated with Monte Carlo to correlate these variations in biological damage with specific physical properties or interactions. Both irradiation and simulation have been performed in water as a representative biological medium since 70% of a cell is composed of this medium.

## 2. Results

### 2.1. Clonogenic Test Reproducibility 

In order to verify the reproducibility of the results, preliminary measurements were carried out with the cells placed at a fixed depth of 24.5 mm ± 0.5 mm and delivering radiation doses of 5 Gy. A total of four irradiation sessions were performed under these conditions. In each session, three samples were irradiated, and the other three were reserved as control references, i.e., not irradiated.

The morphological damage caused by the irradiation is evident when the cells are observed under the microscope, as shown in Figure 1.

Some important aspects of these measurements must be controlled to improve their accuracy and reproducibility. One of these features is the dose uniformity in the cell sample. In our study, the dose profile of the irradiation field (10 cm in diameter) can be considered flat over the extent defined by the flask. In addition, the dimensions of the flask are large enough to minimize the effects of dose variation near the walls [67]. Another experimental aspect to be considered is the presence of backscatter on the cells [68]. In our work, the irradiations were always performed in a water phantom large enough (30 cm × 30 cm × 30 cm) to allow complete backscattering of the cells. The same conditions were used for monitor unit (MU) calculation and the Monte Carlo simulation.

These preliminary measurements yielded a survival fraction (SF) of 0.16 ± 0.03 (1 SD) for 5Gy delivered by a nominal 6 MeV-energy electron beam. The 50% depth of the PDD, R50, is a commonly used energy parameter for electron beams. Since our R50 depth in water was 24.2 mm, the obtained SF value can be considered as a reference value measured at this depth. Therefore, the overall uncertainty assigned to our method (±0.03 over 0.16) is close to 20%.

### 2.2. Dependence of Cell Survival on Water Depth

This section presents the results of the effect of water depth on HaCaT cell survival after 6 MeV electron irradiation.

First, to make the experimental data more comparable, each of the three series of SF measurements was normalized to a value of 15 mm depth, common to all series (±0.3 mm). Once normalized, the SF values were scaled so that a linear fit to all experimental data gave a value of 0.16 at 24.2 mm depth, according to the result presented in the previous section. It is important to note that in this section, we only examine relative changes in biological damage as a function of depth, an aspect related to the slope of the fit.

The range of depths studied varied from 15 to 33 mm, including different clinical areas. Radiation therapy aims to deliver a lethal dose to the target volume while minimizing the dose to healthy organs or tissues. Therefore, variations in biological effectiveness should be studied over a wide range of depths. The first point, at 15 mm, represents the depth at which the absolute dose is typically prescribed for a 6 MeV electron beam to treat the surgical bed immediately above. Beyond this depth, healthy tissue is irradiated. At a depth of 33 mm, approximately 10% of the maximum dose is received.

The final HaCat cell survival fraction results for 18 measurements after delivering 5 Gy at each depth are shown in Figure 2. As can be seen, the trend indicates that the biological damage to cells increases with increasing water depth. SF decreases from 0.22 to 0.10 over the range of measured depths with a slope of −0.007 (ΔSF/mm) according to a linear fit.

A Pearson test on the data yields a correlation coefficient of r = −0.786 (t = 0.00001), indicating a linear correlation between SF and water depth. Our reproducibility study, which was always conducted at a single depth, showed no trend but simply statistical fluctuations. Even if the biological behaviour was not strictly linear, we do not expect significant differences based on our experimental data.

Electrons continuously lose energy as they pass through the medium. This behaviour, typical of charged particles, makes the interaction pattern with the medium clearly dependent on depth. The proportion of low-energy electrons, mainly associated with more significant biological damage, increases with depth. Our experimental results indicate that these physical aspects translate into differences in biological damage for a 6 MeV electron beam in water. This result agrees with those who have measured this dependence with depth [49,50]. 

### 2.3. Monte Carlo Simulation Results

As the Methods section describes, the Monte Carlo simulation was carried out in two steps (see Section 4 for details). The first step simulation was performed within the Geant4 Electromagnetic Physics toolkit framework [69], using the Geant4-based Architecture for Medicine-Oriented Simulations (GAMOS) [70]. By means of this modelling procedure, the energy distribution of the primary beam was determined by considering the beam on the surface of the water phantom as formed by a combination of monoenergetic electron beams in the energy range (0.5–7.5 MeV). The simulated energy spectrum at the surface of the water phantom, shown in Figure 3, was considered as the initial energy distribution of the electron beam for the first step simulation procedure. The relative energy deposition percentage as a function of depth (PDD %) obtained with this energy spectrum and the experimental values are shown in Figure 4 for comparison. As may be seen in this figure, there is an excellent agreement between the simulated and experimental PDD curves.

The Geant4-GAMOS procedure used in this study adopts a “condensed history” technique [71] and applies variance-reduction methods [72,73], which remarkably reduce statistical uncertainties. Using these techniques, we lose information about single collision processes, but the energy deposition and the energy distribution of electrons as a function of depth in liquid water are accurately simulated within an uncertainty of 5%. The simulated energy distributions of the electron beam on the cell layer at the three considered positions (15.3, 24 and 30 mm, respectively) are shown in Figure 5. As expected, the averaged energy of the primary beam decreases with depth while the attenuation of the beam intensity increases. Note that in these conditions, to deliver the same dose to the three considered positions of the target site, the irradiation time must be increased with the depth, which in turn results in the dose rate decreasing with the depth.

The LEPTS results of the number and type of collision events taking place in each of the three positions of the cell layer are shown in Table 1. Note that incident electrons are those of the primary beam reaching the cell layer with the corresponding energy distribution (Figure 5) and those of the low-energy secondary electrons (10 eV on average) formed within the 10 nm thick layer of water in contact with the cell layer. These numbers correspond to the averaged values per incident electron when a constant dose of 5 Gy for each position is delivered in the cell layer.

For each position of interest, the number of primary electrons generated in the simulation, with the energy distribution shown in Figure 5, was high enough to ensure that the energy delivered corresponded to an absorbed dose of 5 Gy with statistical uncertainties better than 1%. In these conditions, the energy per primary electron deposited in the different zones of the sample holdercell layer, methacrylate insulator and ionization chamber, respectively) are presented in Table 2.

We calculated the molecular damage by combining the data available in Table 1 and Table 2 with the number and energy of the electrons required to deliver 5 Gy at the positions of interest, according to the energy distributions and intensities shown in Figure 5. This molecular damage is determined by the total number of molecular dissociations induced by both primary and secondary electrons to the target cells (assumed to be formed of water). Here, we considered that only some inelastic processes are able to dissociate the molecule, namely ionization, electronic excitation, vibrational excitation and electron attachment. Although not all these inelastic processes finally lead to molecular dissociation, as we are considering relative values and the percentage of dissociation to the total number of processes could be similar for each collision type, using the total number can be considered as a good approximation. Elastic processes are needed to properly simulate the single electron tracks but are irrelevant for energy deposition assessments, and rotational excitations are responsible for heating the target but cannot induce molecular dissociations.

The relative biological damage has been directly estimated from the survival fraction values shown in Figure 2. Relative molecular and biological damage, normalized to position 1, are plotted in Figure 6. To simplify the comparison, we used the linear fit described above. As explained in Section 2.1, the estimated uncertainty associated with the SF measurements and the biological damage is within 20–25%. Concerning the simulation, the total uncertainty has been estimated to be about 10% from a quadratic combination of the statistical uncertainty of the Monte Carlo procedure (1%) and the uncertainty limits of the interaction cross sections (7–10%) [74].

## 3. Discussion

HaCat cells are commonly used in biological experiments, but published data on the survival fraction after irradiation are scarce. Our reference SF value of 0.16 is not far from the few published values for HaCat cells and 5 Gy for different radiation beams. In the case of photons, SF values close to 0.10 (0.07–0.13) have usually been reported for ^137^Cs [75,76] and ^60^Co [77]. In the case of Ref. [76], the highest dose delivered was 3 Gy, so the authors obtained an extrapolated value of 5 Gy by a linear fit to the measured data. Meade et al. [78] have measured a survival fraction value as high as 0.24 for the same cells and 5 Gy for ^60^Co irradiation,. For clinical protons, SF values of 0.12–0.14 were reported by Mara et al. [79] but measured in PMMA (polymethyl methacrylate) instead of water. The authors did not find published survival results for HaCat cells irradiated with MeV electrons. 

The experimental uncertainties found in this work can be considered typical for cell survival measurements [80,81]. Although these values may depend on each experiment, cell line, particle, energy or dose range [82], it can be stated that the level of experimental uncertainty is usually high. In addition, although not generalizable due to the presence of different variables, the relative uncertainty tends to increase with the dose (i.e., lower survival rates). In our case, the uncertainty value obtained at a depth of R50 was applied to all other depths. Since we followed the same procedure for each depth and administered the same dose (5 Gy), we did not expect significant differences with the other depths. 

To provide a clinical context of our results, the variation in the measured survival fraction (~0.2–0.1) when delivering 5 Gy at different depths is comparable in magnitude to that obtained with the LQ model [83] by varying the dose from 4 to 5.5 Gy. To derive these LQ-values, we have selected from among the α-β data published for HaCat cells those giving the value closest to our SF for 5 Gy [79] (0.13 vs. 0.16), i.e., α = 0.30 and β = 0.02. Given that reirradiations due to recurrent disease represent a significant proportion of intraoperative electron beam treatments, differences in biological damage, such as those found in this work, may not be irrelevant, especially in healthy organs where the dose received in previous treatments is close to the maximum tolerable dose limit. As already mentioned in the Introduction section of this paper, for the 6 MeV beam, Chattaraj et al. [63] found that using the TDRA theory, RBE varied from 0.74 to 0.84, both values relative to ^60^Co, over the range of water depths in our work (1.5–3 cm). The variation of RBE per mm is 0.0066 for HGS cells, similar to that measured in our study for the HaCat cell survival fraction (0.007 ΔSF/mm).

We have found an excellent agreement between the biological damage derived from our experimental SF values and the simulated molecular damage defined by the number of molecular dissociations induced by primary and secondary electrons. In all cases, the absorbed dose is the same (5 Gy), so the increase in the induced damage with depth should be attributed to the spectral changes of the primary beam and the energy dependence of the interaction cross sections. Another feature that has not been considered here is the dose rate. The simulation launched primary particles one by one with kinetic energies determined by their energy distribution function (Figure 3 and Figure 5) with no specific rate. On the contrary, the dose rate decreases with depth within the experimental conditions. It was typically 10.2, 6 and 2.2 Gy/min for each respective position of the target. The good agreement between our experiment and simulation seems to indicate that, in the present conditions, the dose rate does not substantially affect the induced biological damage. Within the framework of the LQ model, the dose rate effect is associated with the β component, which is related to reparable damage. HaCaT are normal early responding cells with a small β value, so the dose rate dependence is, in theory, of limited importance. However, we should note here that repair mechanisms and long-time processes mainly determine dose-rate effects, so a complete interpretation of these effects would require a detailed analysis of the time evolution of the observed damage. In addition, previous studies [84,85] predicted a general increase in biological damage with the dose rate, in contradiction with the present results. 

Due to the reduced thickness of the cell layer (20 microns), the observed effects are weak, almost competing with the quoted uncertainty limits. In order to confirm this tendency, additional experiments with different target geometries and volumes would be required.

Given the results of our research, it is pertinent to consider other scenarios in which variations in the radiation spectrum may occur. An illustrative example is the use of intensity-modulated radiation therapy (IMRT), in which the fluence of the beam, photons in this case, is modulated through attenuation in the Linac MLC (Multi Leaf Collimator) system. A study by Ezzati et al. [86] investigated the impact of spectrum changes on DNA damage both within the treatment field and in regions beyond using Monte Carlo simulation techniques. However, the study did not observe significant differences in the relative biological effectiveness (RBE) inside and outside the treatment field. It should be noted that photons exhibit different behaviour than electrons, and the alteration of the secondary electron spectrum produced in this case is slight compared to the primary electron beam in our study.

## 4. Materials and Methods

Irradiations were conducted at the Department of Radiation Oncology of the Hospital Ramón y Cajal (Madrid) with an electron LINAC. This model (LIAC HWL, SIORT [87]) has been designed explicitly for intraoperative treatments (IOeRT), providing dose per pulse values 20 times higher than a conventional LINAC, significantly reducing irradiation time. Despite operating at high dose rates, the only shielding required may be the beam absorber, supplied by the company, in order to protect the areas below the beam [88].

The energy range of electron beams is 6–12 MeV, and applicators with diameters ranging from 3 to 12 cm can focus the beam on the tumour while avoiding the surrounding healthy tissue. The 6 MeV-electron beam, the lowest energy beam available, has been used for this study. Preliminary tests established that a dose of 5 Gy was adequate for these cells since it caused appreciable damage without excessively lengthening irradiation times.

Epithelial HaCat [89] cell cultures have been considered the most appropriate for this study. These cells are derived by spontaneous transformation of aneuploid immortal keratinocyte cell lines from adult human skin. HaCaT cells are widely used in scientific research because of their high capacity for differentiation and proliferation in vitro. In addition, they adhere to a surface in the form of a micrometre monolayer, which allows them to be precisely placed at the desired depth. This point is critical since the dose gradient for electron beams can reach up to 10%/mm.

### 4.1. Cell Culture 

Human Keratinocyte Cell Line HaCat was cultured in DMEM with 10% FBS, 1% Pen Strep Glutamine (Gibco ™, Thermo Fisher Scientific, Waltham, MA, USA), and maintained at 37 °C and 5% CO_2._ One day before irradiation, 4 × 10^5^ cells were seeded in Falcon^®^ 12.5 cm^2^ rectangular canted neck cell culture flask with a blue vented screw cap, as shown in Figure 7a. The cells were fixed on the flask’s back wall, which was filled with a culture medium for irradiation. Finally, the lid was sealed with plastic film to prevent water seepage into the cells (see Figure 7b).

### 4.2. Irradiations

The flasks were placed in a 3D water phantom (BluePhantom2, IBA), so the irradiation setup included a layer of cells in a flask filled with biological medium and submerged in liquid water. A plastic holder was designed to fix the flask to the water phantom. This holder also allowed for the placement of an ionization chamber just below the flask in a reproducible and controlled position from the cell layer (Figure 8a).

The irradiations were performed with the 10 cm diameter applicator since this is the reference applicator with which the treatment unit is calibrated. Moreover, this size makes the dose homogeneous in the plane containing the cells. Figure 8b shows the complete irradiation setup, with the applicator flush with the water surface.

A PPC05 plane-parallel ionization chamber (IBA) [90] was used as it is suitable for minimizing the recombination effect associated with the high dose per pulse of the machine [91]. For relative measurements, such as PDDs (percentage depth dose), the electrometer integrated into the BluePhantom2 (IBA) was employed. However, the ionization chamber (IC) was connected to a Max4001 electrometer (Standard Imaging, Middleton, WI, USA) [92] for absolute dose measurements. According to the water phantom user’s guide [93], the positioning servo control accuracy and reproducibility is ±0.1 mm.

This situation leads to a positioning uncertainty of approximately 0.5–1 mm. However, the absolute dose measurement is performed in a low gradient area, so the impact on the dose is minimal. On the contrary, in this study, cells were irradiated at high gradient depths (up to 10%/mm), so a more precise knowledge of the depth of the cells was necessary. 

The actual depth at each irradiation was calculated from the IC measurements and the distance from the IC reference point to the cell layer. For this purpose, the flask was examined using a CT scanner, revealing wall thicknesses of 1.8 mm and a separation of 21 mm between them, as depicted in Figure 9.

The distance between the measurement point (IC reference point) and the cells was 3.8 mm for all irradiations: 1.8 mm from the thickness of the flask back wall, 1 mm from the IC cap (in contact with the flask surface) and 1 mm for the distance from the reference point to the outer surface of the IC.

The ideal scenario would be to irradiate cells in liquid water. Since this is not possible, the effect of the flask was first studied in terms of absorbed dose. For this purpose, percentage depth dose profiles (PDD) were acquired for 6–12 MeV energies with a flask filled with cell culture medium and compared with those clinical PDDs in the absence of the flask. Thus, the effect of both the flask and cell medium on PDD could be determined. As seen in Figure 10a, expected small changes in the spectrum due to the use of the flask do not affect the PDDs, including the lowest available energy, which is 6 MeV. In these benchmarking tests, the minimum depth was 27 mm, the distance from the filled flask surface, flush with the water surface, to the IC reference point. 

Figure 10b shows the radial dose profiles in water at a depth of 9 mm for the 10 cm applicator and 6 MeV. The PDD and radial profiles at different depths form the 3D absorbed dose map.

The following procedure was designed to ensure a dose *D* to cells located at a depth *d*:i.The cGy/MU ratio is measured at 9 mm, the maximum PDD for 6 MeV. This measurement is performed without a flask and is used both to know the absolute calibration of the unit at the time of irradiation and the ionization charge at 9 mm.ii.The cells are attempted to be placed at the planned depth.iii.A total of 300 Monitor Units (MU) are delivered.iv.The depth of the IC reference point is then calculated with the clinical PDI (percent depth ionization) profiles through the ratio of this charge to the one previously obtained at 9 mm. Notably, electron PDD and PDI measured with an ionization chamber differ due to changes in the ratio of the mass-electronic stopping powers from water to air with depth [94]. The cells were always placed 3.8 mm above the IC reference point.v.Once the actual depth of the cells is known, the remaining MUs to reach *D* are calculated from cGy/MU ratio and the PDD.

A total of 1 MU is defined as a given amount of charge (nC) produced in the LINAC monitoring ionization chamber to deliver 1 cGy in water at a given depth (9 mm for the 10 cm applicator with an energy of 6 MeV). The equivalent of the MU in a natural radiation emitter is time.

The irradiations were grouped into two sets according to two different objectives. On the one hand, the reproducibility of the overall procedure was studied in four irradiation sessions. For this purpose, 5 Gy was administered to three flasks in each session, and three were left unirradiated. The cells were always at the same nominal depth of 24 mm. On the other hand, several irradiation sessions were performed to study the observed biological damage as a function of water depth. In each session, six flasks were irradiated with 5 Gy at different depths ranging from 15 mm to 33 mm, always leaving one flask unirradiated as a control. A relationship was previously established between the volume of cell medium (mL) and the depth of the cells in the flask to irradiate cells at depths of less than 23 mm.

In all cases, the procedure described above was followed for each irradiation, making the necessary corrections in the MU calculation to consider the actual depth.

### 4.3. Cell Analysis and Survival Fraction

One day after each irradiation session, a clonogenic assay was conducted to study cell biological damage. Irradiated cells were first trypsinised, and then alive cells were counted by Trypan blue exclusion (Sigma-Aldrich, Burlington, MA, USA) using the counter TC20 (Bio-Rad, Hercules, CA, USA). Cells were seeded at 1 × 10^3^ cells/well in six-well plates. Approximately eight days later, colonies were fixed in methanol for 10 min, stained for 30 min with Giemsa 0.02% (Sigma-Aldrich) and washed twice with water until Giemsa residue was removed. Finally, images, as shown in Figure 11, were captured using ChemiDoc^TM^ (Bio-Rad).

The percentage of inhibition relative to the control was calculated to obtain the SF value. These calculations were performed automatically by employing the *ColonyArea* plugin [95] for *ImageJ*. SF was estimated from the *colony area parameter* (%) ratio of irradiated cells to the control, where the *colony area parameter* (%) is defined as the ratio of the number of pixels above a threshold value to the total pixels. Figure 12 shows the final appearance of a well with *ColonyArea* automatic colony detection.

### 4.4. Monte Carlo Simulation

Monte Carlo (MC) methods have been used for decades to simulate radiation transport in different media. They have been extensively used in biomedical radiation applications to calculate the energy deposition and absorbed dose in simulated biological media [96]. Basically, MC procedures are based on random number generators, which, by means of multiple reiterations of possible collisions of the radiation particles with the atoms and molecules constituting the medium, are able to obtain the most probable particle tracks according to the probability distribution functions used as input data. Using big computers, a large number of tracks can be sampled to provide reliable data on the energy deposited in the different areas of interest. The statistical uncertainty of this sampling procedure can be as good as the computing time invested in it. However, the kind of output information and the actual uncertainty of these results directly depend on the type of collision processes considered in the simulation and the accuracy of the probability distribution functions associated with these processes. In addition, due to the high energy of the radiation particles (photons, electrons or ions) making radiotherapy beams, the number of possible collision processes is huge, and the sum of their associated single uncertainties can lead to considerable errors in the output data. There is no superior MC simulation procedure in these conditions, but the most appropriate (or a combination of them) must be chosen for each application. In this study, the primary radiation beam is formed by high-energy electrons (typically around 6 MeV), and our target medium mainly consists of water. Note that a primary 6 MeV electron can suffer about 2 × 10^5^ inelastic collisions with water molecules along the slowing-down track until its final thermalization. Around 80% of these inelastic collisions ionize the water molecule, generating secondary electrons that can also collide with the target molecules. In the case of electrons, elastic processes are not substantially contributing to the energy deposition. However, they produce electron deflections that contribute to the configuration of the electron track and, therefore, need to be considered in a realistic track simulation procedure (this means at least 10^5^ additional processes to consider). In these conditions, we had to abandon the idea of using a detailed event-by-event Monte Carlo simulation procedure accounting for the elastic and all the possible inelastic scattering (ionization, electronic, vibrational and rotational excitation, electron attachment) of 6 MeV electrons slowing down in water.

General purpose Monte Carlo methods are mainly focused on the energy deposition of high-energy particles (photons, electrons and ions) in different media. They reduce the number of collisions to be considered in the simulation by employing a “condensed history” procedure. In the case of electrons, the “standard” electromagnetic package of the Geant4 toolkit is commonly used to simulate their interactions with matter to provide information on the energy deposition, stopping power and ranges, which generally agree with the well-established NIST database. We can then assume that the energy distribution of the beam at different depths given by this simulation is also in agreement with these reference data. In particular, for medical applications, the GAMOS interface is a good complement to Geant4 by incorporating useful tools to model the geometry and properties of the materials used in biomedical applications of radiation. However, in favour of improving the statistical uncertainty and reducing computation times, these methods lose information about single events. For applications requiring a detailed description of the number and type of collision events occurring in the target area, an event-by-event MC simulation procedure must be used (see Ref. [96] for details). The counterpart is that considering single collision events notably increases the number of processes, thus increasing the simulation time and cumulative errors.

The geometry of the present simulation is shown in Figure 13a. We have considered three different positions of the flask containing the cell layer. Figure 13b displays the PDD curve as measured with an ionization chamber for a “nominal” 6MeV electron beam, indicating the depth of these positions. According to the above considerations, to solve this particular problem, we must distinguish three different regions into the target volume: the surrounding water, the thin water layer in contact with the cells (10 nm thick) and the cell layer (20 microns thick) surface. We do not need details on the single scattering events for the surrounding water region, and energy deposition and energy distribution of the beam would be enough. In this region, a Geant4-GAMOS simulation will be the most reliable to provide that required information. However, information about the number and type of interactions are also needed in the layer in contact with the cells (where generated secondary electrons can reach the cell surface) and the proper cell layer in order to correlate these data with the observed biological damage.

We then decided to use our Low Energy Particle Track simulation (LEPTS) code to model the electron interactions in the mentioned water and cell layers, assuming that both are composed of pure water. The LEPTS procedure has been described in previous articles [97,98,99], so we will refer only to details directly connected with the present results. It is an event-by-event Monte Carlo simulation using a complete set of interaction probabilities (cross sections) and angular and energy loss distribution functions as input data [100]. From a technical point of view, this is a standard simulation procedure. However, its strongest point is the input data set on which it is based. Cross sections, angular and energy distribution functions are derived from a critical and detailed compilation of all the data available in the literature, incorporating new measurements and calculations where required, especially for the lower energies [101], and checking their consistency through specific validating experiments [74]. This data set extends over the energy range (0–10 keV). Above this energy range, where the First Born approximation applies, input data are supplemented with the standard Livermore cross-section library [102]. Some information about the input data is briefly summarized in the following subsubsections.

#### 4.4.1. Interaction Probabilities

These data are directly derived from the integral elastic (IECS) and integral inelastic (IICS) cross-sections. IECS values used in this simulation come 100% from calculations. In fact, it is impossible to measure IECSs directly, but they are commonly derived from integrating differential cross-section (DCS) measurements, which requires some additional theoretical approaches [97]. To produce the IECS, we used our independent atom model (IAM) calculation complemented with the screening corrected additivity rule (SCAR), considering interference effects (IAM-SCAR+I) (see [98] and references therein for details). This procedure has been proven to be reliable within 10% for impact electron energies ranging from 10 eV to 10 keV for a variety of molecular targets [98,99,100]. We should note here that we consider elastic processes only those in which no energy is transferred to excite any internal degree of freedom of the target molecule, i.e., when the total kinetic energy of the projectile-target molecule system remains constant under the collision. Below 10 eV, more sophisticated approaches considering molecular orbitals and using “ab-initio” methods to account for electron correlations have been considered. In particular, we included the R-matrix [74] calculation for water [102], which showed remarkable consistency with the IAM-SCAR+I calculations for different molecular targets [98,99,100].

IICS is formed by the sum of the contributions of all the inelastic channels that are open (energetically accessible) at a given incident energy. For increasing excitation energies, the first inelastic open channel corresponds to rotational excitation processes (0.012 eV average excitation energy at 300 K [103]). Vibrational excitation, electronic excitation and ionization of water molecules have their threshold energies at 0.2 eV [104], 4.5 eV [105] and 12.8 eV [106], respectively. Condensation effects tend to lower the ionization energy [105,106]; hence we assumed an extrapolated ionization energy of 10.79 eV for the liquid water. In these conditions, our IAM-SCAR+I calculation provided the reference data from which most of the IICS values are derived. However, they sometimes needed to be complemented with available theoretical or experimental data. The selected input cross-section sources for the 0–10 keV impact energy range can be summarized for each considered inelastic channel as follows:Rotational excitation cross-sections are calculated by assuming that the molecule behaves as a free rotor and making use of the First Born approximation [107,108].Vibrational excitation cross-section data come from a detailed compilation of the experimental data available in the literature as described in [100]. The overall experimental uncertainties assigned to these data are within 20–25%.We measured ionization cross-sections by combining a transmission beam apparatus with a time-of-flight mass spectrometer [108]. Typical uncertainties assigned to these experimental data are within 7–10% [107,109].Electronic excitation cross-sections are taken from the literature [109]. Having theoretical or experimental data on all electronic excited states available at each incident energy is almost impossible, even with such an intensively studied molecule like water. As explained in [100], we compiled individual electronic excited state cross-sections from previous experimental studies, which were validated by comparison with recommended data from review articles such as that by Itikawa and Mason [110].Neutral dissociation cross sections. As explained in Ref. [100], the difference between our calculated total inelastic cross section and the sum of the cross sections corresponding to the ionization, vibrational and electronic excitations, i.e., the “remaining” inelastic channels, are considered to be neutral dissociation processes. Direct measurements of neutral dissociation cross-sections are very complicated and thus scarce. However, in a previous study, we showed that this assumption is compatible with the few measurements available in the literature [111,112].Inner-shell excitation and ionization. The probability of exciting and ionizing the k-shell of the oxygen atom forming the water molecule has been estimated from the energy loss spectra by correlating the total electron intensity corresponding to these processes with that corresponding to the molecular excitation and ionization processes. Although the cross-section for inner-shell excitation and ionization is about two orders of magnitude less than that corresponding to the valence shell, its average energy loss (around 535 eV [112]) is more than one order of magnitude higher than the average energy of the outermost electron shell (around 35 eV [107]). Therefore, they must be considered in the simulation procedure.Multiple ionization cross sections. This is an inelastic channel recently incorporated into our modelling procedure. In our previous simulations using LEPTS, we assumed that the double ionization of the water molecules was irrelevant due to the low cross-section we may expect for this process. However, a recent calculation from Champion et al. [113] revealed that the energy transferred in this type of process is several orders of magnitude higher than that in single ionization and that it increases with the incident energy. Therefore, it is important in terms of energy deposition. In fact, some discrepancies we showed between our simulated electron stopping power in water [107] and that given on the NIST webpage [114] for energies above 1 keV disappear when this process is included in the simulation.Electron attachment cross-section. This is a resonant process whose cross-sections have been accurately measured in several experiments compiled by Itikawa and Mason [110]. We have taken the recommended cross-section values from Ref. [109].Thermalization. Electrons with energy below 25 meV are considered as thermalized electrons in equilibrium with the medium.

The scattering cross sections used to include all the above scattering processes in the present simulation are plotted in Figure 14.

#### 4.4.2. Angular Distribution Function

An important feature of an event-by-event Monte Carlo simulation procedure is sampling the angular distribution of scattered electrons for every single collision. We will distinguish here between elastic and inelastic collision processes.

The calculated differential cross-sections for elastic scattering are normalized to constitute the angular distribution probabilities for each incident energy. The sampling procedure is more complicated for inelastic scattering processes since this angular distribution depends on the energy transferred to the target. In addition, the energy and angular distribution of the ejected electron are also needed for ionizing collisions. Theoretically, we could derive this information from the double (DDCS) and triple (TDCS) differential cross-sections. In practice, this information is not available for all the required energies, and the full integration of the available DDCS data is not always consistent with the measured total integral inelastic cross sections. For this reason, we proposed a relatively simple empirical formula [100] to derive the angular distribution of the inelastically scattered electrons as a function of the angular distribution corresponding to the elastic scattering and the transferred energy. This formula is the result of systematic energy loss spectrum measurements for different scattering angles and can be written as:(1)$$\frac{d^{2}\sigma \left(E\right)}{d\Omega \Delta E}\propto {\left(\frac{d\sigma \left(E\right)}{d\Omega }\right)}_{el}^{1-\Delta E/E}{\;}^{\;}$$
where ${\left(\frac{d\sigma \left(E\right)}{d\Omega }\right)}_{el}^{\;}$ is the angular distribution of elastically scattered electrons for the same incident energy, and ΔE is the energy loss during the collision. For ionizing collisions, the energy of the ejected electron is given by ΔE-IP, where IP is the ionization potential, and its angular distribution is derived according to the energy and momentum conservation laws.

#### 4.4.3. Energy Loss Distribution Functions

The kinetic energy transferred in elastic processes is simply derived from the projectile to target molecule mass ratio and the scattering angle. For inelastic processes, we followed different procedures depending on the inelastic channel considered. For rotational excitation, we assumed a constant energy loss, which is the averaged rotational excitation energy at a temperature of 300 K, i.e., 0.015 eV. For vibrational excitation, electronic excitation and ionization (valence and inner shell) processes, the energy loss distribution functions have been derived from the experimental energy loss spectra. For double ionization processes, we adopted the energy loss energies calculated by Champion [113]. The energy loss distribution functions used in this simulation are plotted in Figure 15 for the most representative inelastic channels.

#### 4.4.4. Simulation Procedure

Another important data required to run the simulation are the energy distribution of the primary electron beam. We mentioned we used a “nominal” 6 MeV electron beam, as designated by the manufacturer of the LIAC HWL accelerator [87]. However, due to the system’s electron accelerating and transport procedure, the electron beam is not monoenergetic but rather presents an energy distribution with a peak energy close to the nominal value, which is 6 MeV in this case. Unfortunately, direct measurements of the primary beam spectra are not available. The fluence of the beam is too high for standard electron spectrometers, and reducing the electron current to the required detection rates would modify the spectrum we wanted to determine. Since the reference parameter for this experiment is the absorbed dose, we followed an alternative procedure to determine the effective energy distribution. We decomposed the beam profile into monoenergetic beams whose intensities were adjusted to provide the same absorbed doses at the depth points of interest (15.3, 24 and 30 mm) as those measured by an ionization chamber placed at these points. The result of this fitting procedure is shown in Figure 3. As can be seen in this figure, the main peak has a slightly higher energy (around 6.5 MeV) with a broad lower energy feature around 4 MeV. We assumed that this is the energy distribution of the primary beam on the surface of the water phantom. The flask, which contains the live cell layer and is equipped with an attached ionization chamber, is sequentially positioned at three distinct points of interest: 15.3, 24 and 30 mm. These depth values are measured from the water surface along the vertical axis of the water phantom (see Figure 13) These depth values are measured from the water surface along the vertical axis of the water phantom (see Figure 13).

As already mentioned, the simulation procedure was performed in two steps for each position of the living cell target. Within the first step, the electron beam transport in liquid water has been simulated by using the standard electromagnetic package of the Geant4 toolkit. This is a condensed history approach simulation procedure; thus, no information on single collision events is given, but it provides reliable information on energy deposition and energy distribution of the electron beam in the considered target volume. This simulation stops near the cell layer (within the 10 nm thick water layer in contact with the cells where generated secondary electrons are able to reach the cell layer) when the absorbed dose in the target area reaches the fixed value of 5 Gy with statistical uncertainties within ±5%. The second step of the simulation starts in this water layer close to the cell layer and is carried out with our LEPTS procedure. As aforementioned, this is an event-by-event Monte Carlo simulation procedure using the interaction probabilities and distribution functions mentioned above as input data. For each of the three considered positions, the energy distribution and the total intensity attenuation of the primary beam are provided by the simulation of the first step.

## 5. Conclusions

We have presented a comparative study of the observed biological damage produced by a clinical 6 MeV electron beam for intraoperative radiotherapy to epithelial HaCaT cell cultures as a function of depth in liquid water. Three representative positions at 15.3, 24 and 30 mm depth, respectively, have been chosen for the simulation, while the experiment has been repeated at different depths ranging from 15 to 33 mm. In all the cases, the delivered dose was calculated to ensure a constant dose of 5 Gy on the target. The biological damage has been estimated from the measured survival fraction through a careful analysis of the colony formation after the irradiation. The molecular damage has been linked to the number of dissociative inelastic processes taking place in the target. This parameter has been derived from the simulated number of processes per incident electron, the averaged energy per electron deposited in the target area and the number of electrons required to reach a constant absorbed dose of 5 Gy. Despite the weak variation of the biological and molecular damage that we found with the 15–33 mm depth range, the agreement between the experiment and simulation has been excellent. This indicates that qualitatively biological and molecular damage increases with depth in liquid water due to the degradation of the beam spectrum towards the lower energies and the increase in the electron scattering cross section for decreasing energies. However, additional experiments for different target geometries would be needed to confirm this tendency and properly quantify this effect.

## Figures and Tables

**Figure 1 ijms-24-10816-f001:**
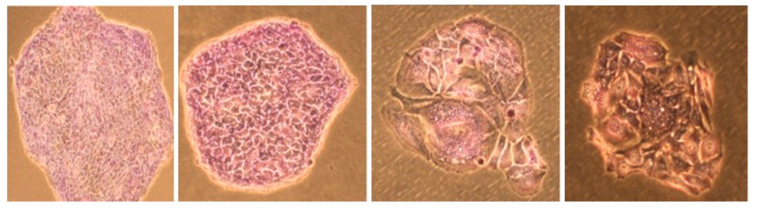
The two images on the left belong to non-irradiated cells, while the two on the right have received 5 Gy.

**Figure 2 ijms-24-10816-f002:**
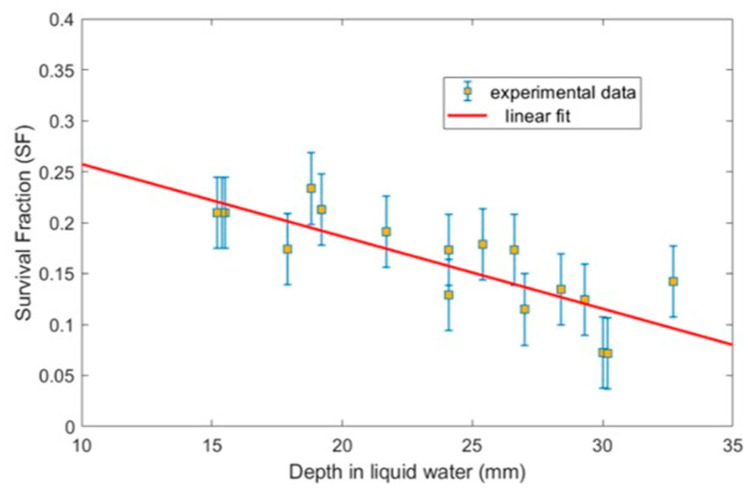
Results of the clonogenic study of HaCat cells with varying depth. Error bars come from the reproducibility study (Section 2.1).

**Figure 3 ijms-24-10816-f003:**
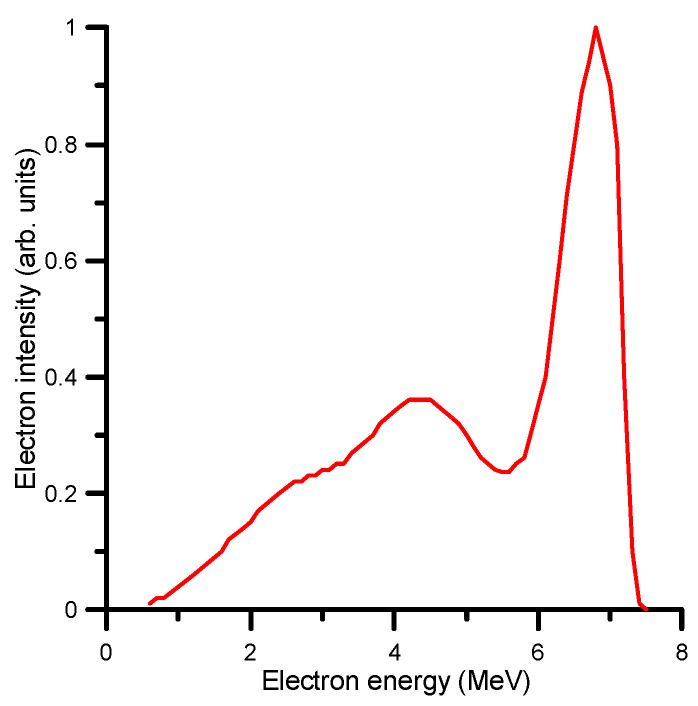
Simulated energy distribution of the primary beam on the surface of the water phantom.

**Figure 4 ijms-24-10816-f004:**
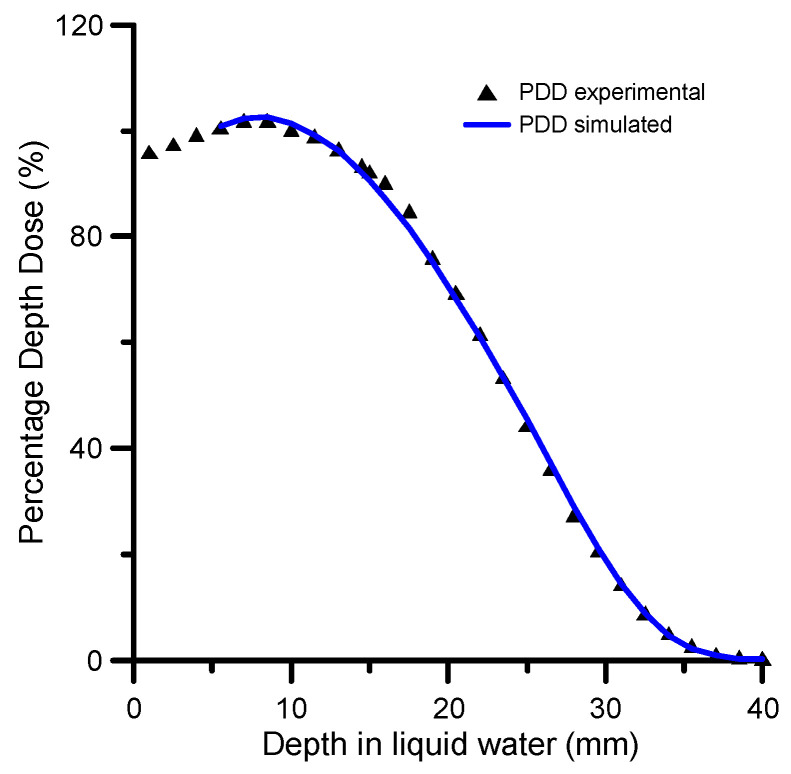
Experimental and simulated 6MeV PDDs.

**Figure 5 ijms-24-10816-f005:**
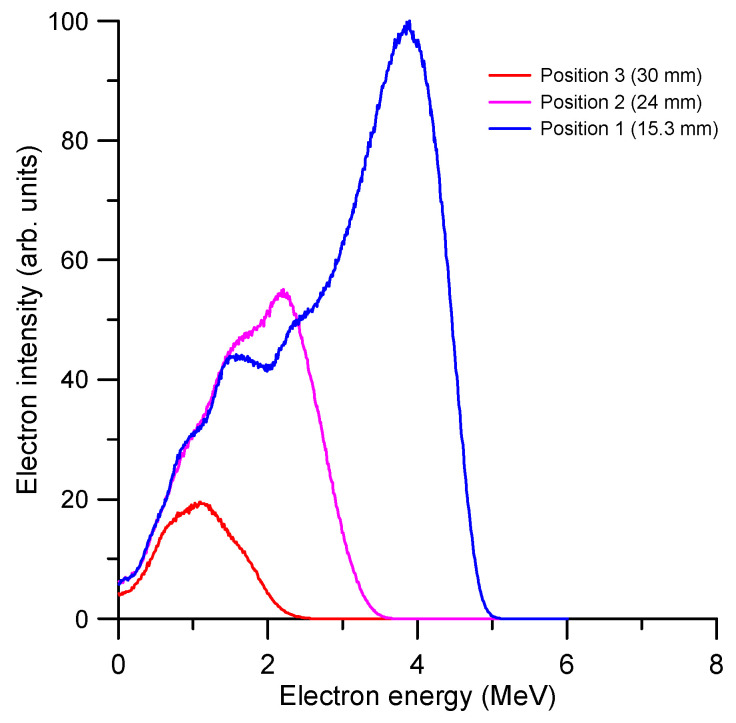
The Geant4.GAMOS simulated energy distributions of the electron beam on the cell layer at 15.3, 24 and 30 mm.

**Figure 6 ijms-24-10816-f006:**
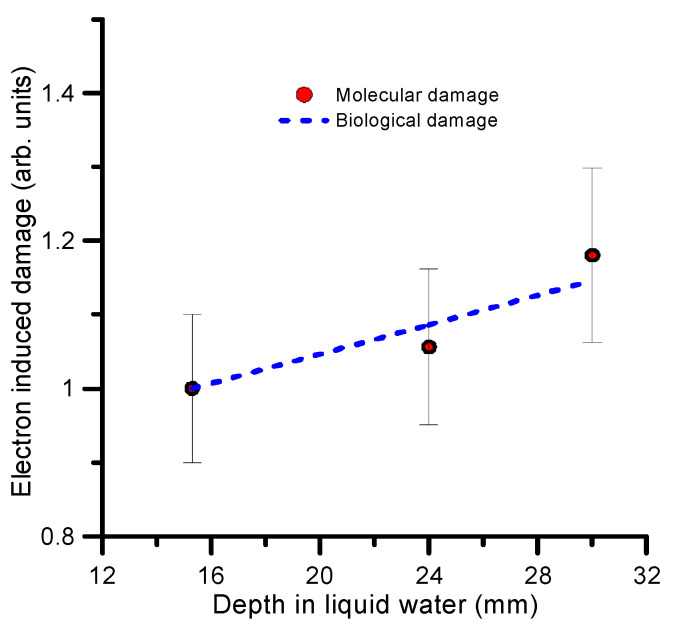
Comparison between the relative biological damage derived from the experimental survival fraction (**- - -**) and the simulated molecular damage (**●**), both normalized to position 1 (15.3 mm). See text for details.

**Figure 7 ijms-24-10816-f007:**
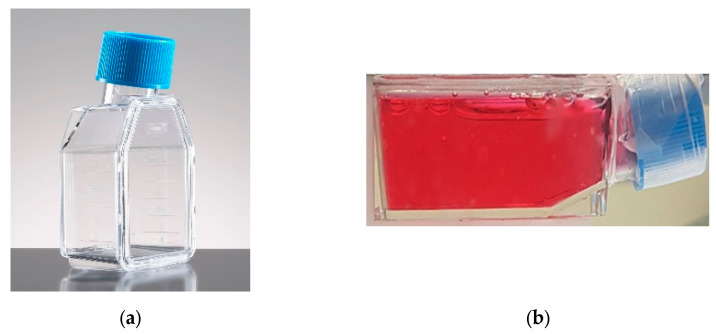
(**a**) A 12.5 cm^2^ culture flask, as the ones used; (**b**) the same flask containing the cell layer and the culture medium, ready for irradiation.

**Figure 8 ijms-24-10816-f008:**
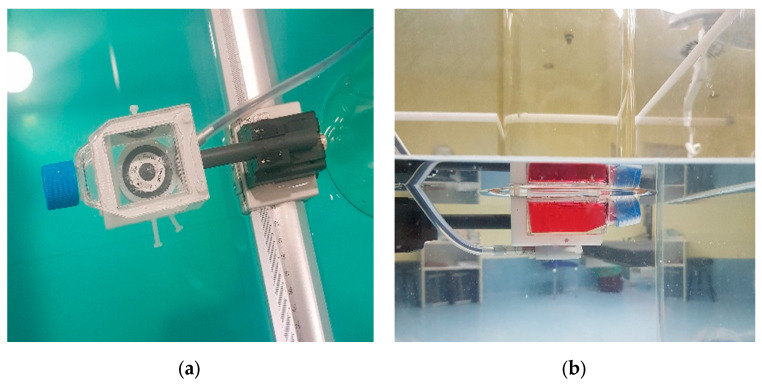
(**a**) Holder containing the flask and the ionization chamber. (**b**) Final setup with the applicator, the flask, the cells and the ionization chamber.

**Figure 9 ijms-24-10816-f009:**
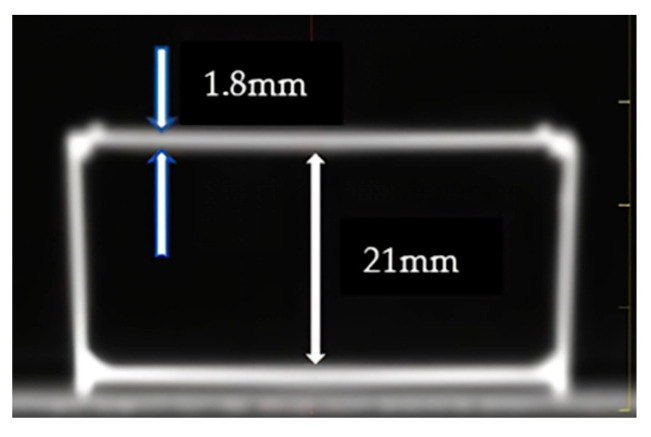
CT slice with the relevant dimensions of the flask.

**Figure 10 ijms-24-10816-f010:**
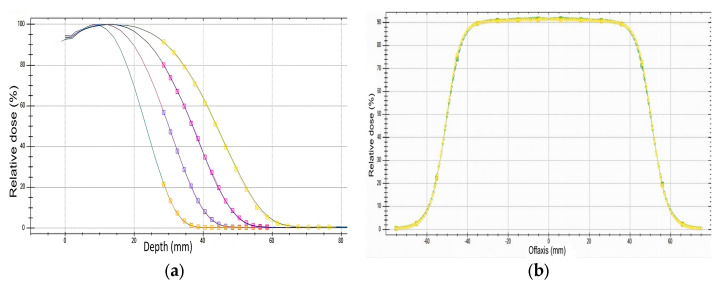
(**a**) The solid lines are clinical PDDs for 6, 8, 10 and 12 MeV without a flask. The dotted lines are PDDs with the flask filled with cellular medium. (**b**) Radial dose profiles at a depth of 9 mm for the 10 cm applicator and 6 MeV.

**Figure 11 ijms-24-10816-f011:**
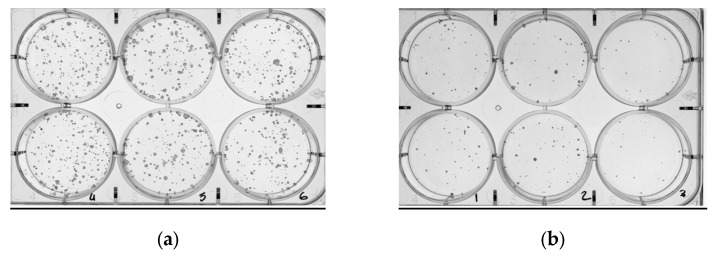
(**a**) Colonies formed for nonirradiated cells. (**b**) Colonies formed after irradiating the cells with 5Gy.

**Figure 12 ijms-24-10816-f012:**
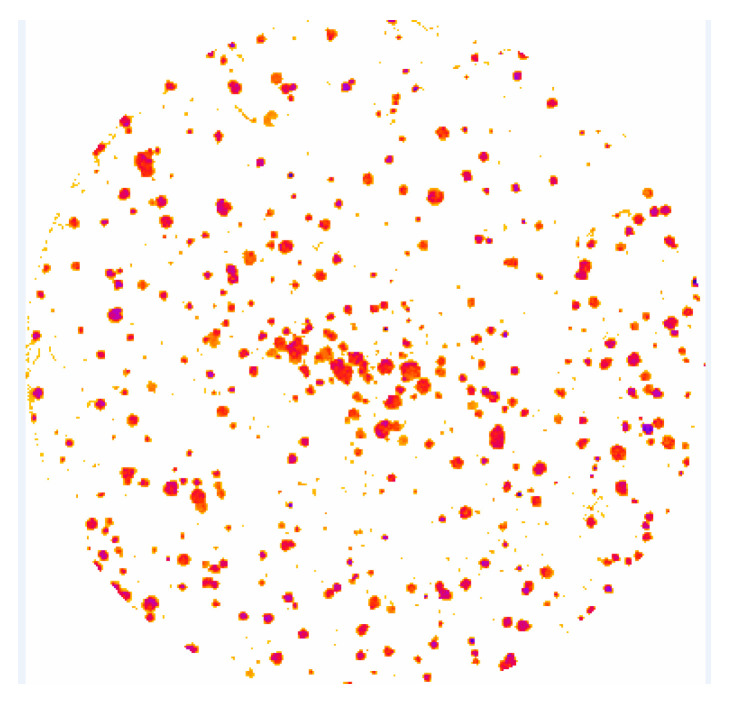
Extracted image by *ColonyArea* software with cell colonies.

**Figure 13 ijms-24-10816-f013:**
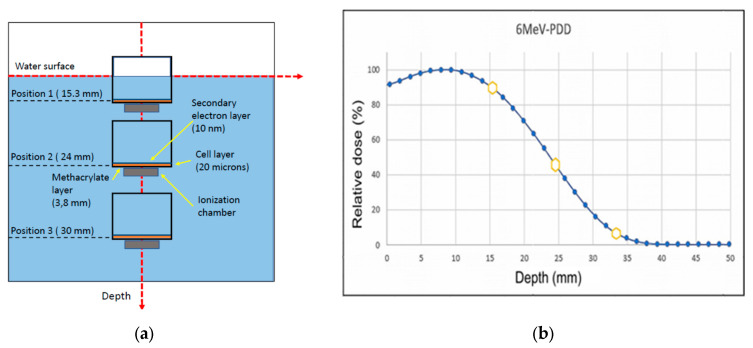
(**a**) Schematic diagram of the geometry assumed for the Monte Carlo simulation. (**b**) Measured 6MeV-PDD indicating the studied depths (
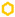
) with Monte Carlo.

**Figure 14 ijms-24-10816-f014:**
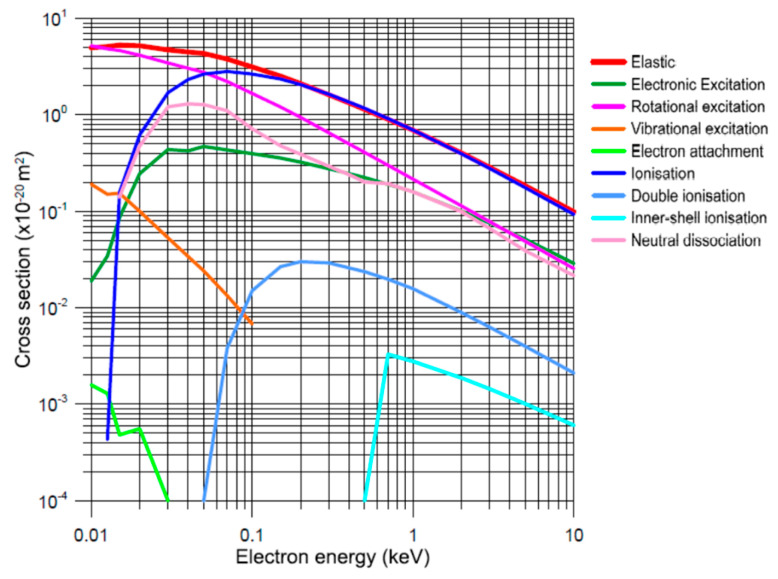
Electron scattering cross sections used for the LEPTS simulation for incident electron energies below 10 keV.

**Figure 15 ijms-24-10816-f015:**
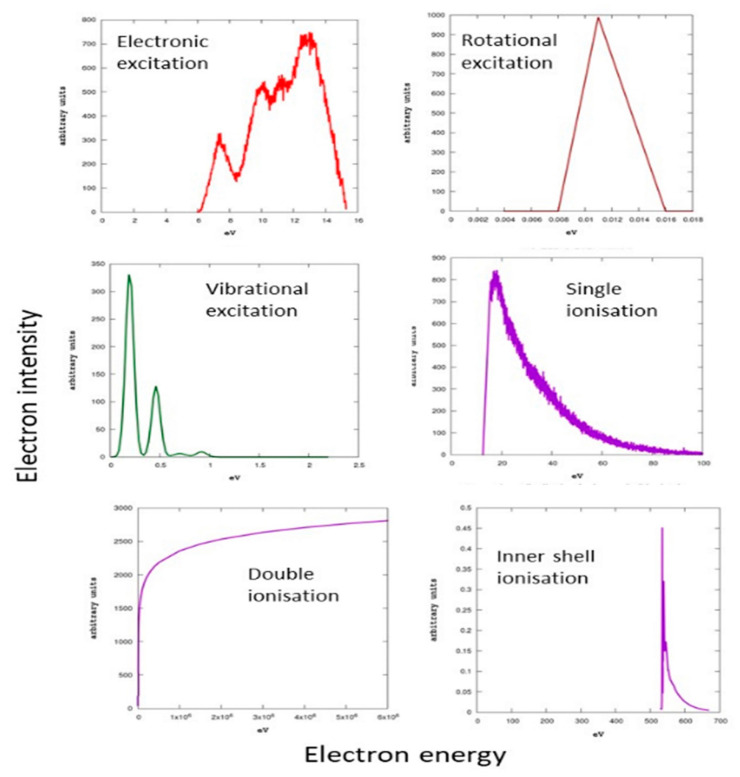
Energy loss distribution functions for different electron scattering processes.

**Table 1 ijms-24-10816-t001:** Averaged number of collisions per primary electron taking place in the cell layer at 15.3, 24 and 30 mm depth for each type of interaction.

Type of Collision Process	Position 1 (15.3 mm)	Position 2 (24 mm)	Position 3 (30 mm)
Elastic	13,567.7	13,986.3	15,251.7
Rotational excitation	24,293.9	25,036.6	27,303.6
Vibrational excitation	402.455	414.776	451.836
Electron attachment	5.16375	5.28861	5.76576
Electronic excitation	30.3162	31.4765	34.5862
Single ionization	77.8115	80.0472	87.5106
Double ionisation	0.947292	1.00743	1.09097
K-shell ionization	4.74 × 10^−1^	4.66 × 10^−1^	4.81 × 10^−1^
Final thermalization	72.7569	74.8771	81.8202

**Table 2 ijms-24-10816-t002:** Averaged energy deposition (MeV), per primary electron, in each one of the regions of the sample holder (see Section 4).

Zone	Position 1 (15.3 mm)	Position 2 (24 mm)	Position 3 (30 mm)
Cell layer (20 microns)	0.0039	0.0040	0.0041
Insulator (3.8 mm)	0.744	0.737	0.713
Ionization chamber	2.16	1.06	0.37
TOTAL	2.91	1.80	1.09

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
