# Peer review of "Dependence of Induced Biological Damage on the Energy Distribution and Intensity of Clinical Intra-Operative Radiotherapy Electron Beams"

_ijms, 2023, doi:10.3390/ijms241310816_

Round 1

Reviewer 1 Report

The paper has well-designed research methods, appropriate statistical analysis and a relatively good interpretation of the results.

The quality of the language in the context of grammar, syntax and spelling is solid, however, when reading the paper I occasionally get a certain impression of confusion.

About the Title of the article, I suggest you to modify it and add the type of article.

Please be sure to use only keywords accordingly to medical subject headings (Mesh word) for a better indexing.

The introduction section is very short and is needed to add other references to increase the quality of the manuscript [10.3390/medicina59020410];[10.1111/joor.13472]

-I suggest you add a table with the list of abbreviations used in the text.

- The materials and methods section is well structured and in line with the results.

- The Conclusion briefly exposes the main findings of the study, underlying the clinical relevance of this study

Punctuation errors are present, I suggest editing them

Reviewer 2 Report

Dear authors,

The paper is well-written and the topic is interesting. The authors have put lots of effort into drafting this manuscript. My only concern is regarding the introduction section, which is quite lengthy.

The paper could go for publication

Reviewer 3 Report

1.  Radiation field and protection requirements?

2. Intensity-modulated radiotherapy (IMRT) techniques?

3. Dose distributions for the first draft of a foils-collimator-applicator system?  

1.  Radiation field and protection requirements?

2. Intensity-modulated radiotherapy (IMRT) techniques?

3. Dose distributions for the first draft of a foils-collimator-applicator system?  
